# *LeGRXS14* Reduces Salt Stress Tolerance in *Arabidopsis thaliana*

**DOI:** 10.3390/plants12122320

**Published:** 2023-06-15

**Authors:** Lulu Liu, Xiaofei Li, Mengke Su, Jiaping Shi, Qing Zhang, Xunyan Liu

**Affiliations:** 1College of Life Sciences and Oceanography, Shenzhen University, Shenzhen 518060, China; liululu124@hotmail.com; 2College of Physics and Optoelectronic Engineering, Shenzhen University, Shenzhen 518060, China; 3College of Life and Environmental Sciences, Hangzhou Normal University, Hangzhou 310030, China; lxf17633576763@163.com (X.L.); smk155321@163.com (M.S.); sjp12302000@163.com (J.S.); zq15025721179@126.com (Q.Z.)

**Keywords:** salt stress, GRXs, *LeGRXS14*, overexpression

## Abstract

Salt stress represents a significant abiotic stressor for plants and poses a severe threat to agricultural productivity. Glutaredoxins (GRXs) are small disulfide reductases that can scavenge cellular reactive oxygen species and are crucial for plant growth and development, particularly under stressful circumstances. Although CGFS-type GRXs were found to be involved in various abiotic stresses, the intrinsic mechanism mediated by LeGRXS14, a tomato (*Lycopersicon esculentum* Mill.) CGFS-type GRX, is not yet fully understood. We discovered that LeGRXS14 is relatively conserved at the N-terminus and exhibits an increase in expression level under salt and osmotic stress conditions in tomatoes. The expression levels of *LeGRXS14* in response to osmotic stress peaked relatively rapidly at 30 min, while the response to salt stress only peaked at 6 h. We constructed *LeGRXS14* overexpression *Arabidopsis thaliana* (OE) lines and confirmed that LeGRXS14 is located on the plasma membrane, nucleus, and chloroplasts. In comparison to the wild-type Col-0 (WT), the OE lines displayed greater sensitivity to salt stress, resulting in a profound inhibition of root growth under the same conditions. Analysis of the mRNA levels of the WT and OE lines revealed that salt stress-related factors, such as *ZAT12*, *SOS3*, and *NHX6,* were downregulated. Based on our research, it can be concluded that *LeGRXS14* plays a significant role in plant tolerance to salt. However, our findings also suggest that *LeGRXS14* may act as a negative regulator in this process by exacerbating Na^+^ toxicity and the resulting oxidative stress.

## 1. Introduction

Soil salinity is an increasingly severe environmental issue in agriculture worldwide [1,2]. Sodium chloride (NaCl) is the key salt accountable for salinization, which can hinder the growth and development of plants. Salt stress causes osmotic stress and ionic toxicity, which decrease turgor pressure and hasten stomatal closure. In addition, in metabolically active intracellular compartments, the excessive accumulation of Na^+^ and Cl^−^ can affect enzyme activities and result in physiological dysfunction. Salt stress also triggers an excess of reactive oxygen species (ROS) in plants, such as singlet oxygen (^1^O_2_), superoxide anions (O_2_^·−^), hydrogen peroxide (H_2_O_2_), and hydroxyl radicals (HO^·^), which act on DNA, lipids, and proteins, leading to irreversible oxidation events and oxidative damage and ultimately leading to cell death [3].

To maintain low ROS concentrations in cells, plants have evolved a variety of precise oxidative scavenging systems to combat oxidative damage. An array of enzymatic and non-enzymatic antioxidants can scavenge ROS, including ascorbate peroxidases (APXs), superoxide dismutases (SODs), catalases (CATs), glutathione peroxidases (GPXs), ascorbic acid, carotenoids, glutathione (GSH), flavonoids, and α-tocopherol [4]. Additional proteins have been identified as playing a role in the signaling of reactive oxygen species within plants, such as glutaredoxins (GRXs). These are small, ubiquitous oxidoreductase enzymes belonging to the thioredoxin (TRX) superfamily that were initially discovered in *Escherichia coli* in 1976 [5,6,7,8]. GRXs can regulate the redox state of thiol groups in cells or utilize the reducing power of glutathione (GSH) to exchange a glutathionylated moiety with target proteins. The reduction of dehydroascorbate (DHA), peroxiredoxin, and methionine sulfoxide reductase by glutaredoxins (GRXs) helps to remove ROS and repair oxidative damage to macromolecules [9,10,11,12,13]. 

Based on the motif present at their active sites, GRXs can be categorized into three distinct classes: CPYC-type, CGFS-type, and CC-type. CPYC-type GRXs can be found in nearly all prokaryotes and eukaryotes, whereas CC-type GRXs are only found in land plants. CGFS-type GRXs contain single-domain and multi-domain GRXs. Single-domain GRXs are similar to CPYC-type GRXs, and multi-domain GRXs occur exclusively in eukaryotes [14,15]. CPYC-type GRXs are involved in regulating the redox state of thiol groups in proteins or small compounds, while CC-type GRXs are mainly involved in developmental processes and stress responses [16,17]. Tomatoes are an important economic crop, and CGFS-type GRXs in tomatoes exhibit similar mechanisms in the plant stress response as CC-type GRXs. For example, the mutants *legrxS14* and *legrxs17* show hypersensitivity to heat and chilling stress, and *legrxS14* mutants exhibit higher tolerance to heavy metal toxicity and drought stress [18]. Limited research exists on the involvement of *LeGRXS14* in conferring salt tolerance; therefore, we examined the phenotype and molecular mechanism of *LeGRXS14* under salt stress in overexpressing plants.

To investigate how *LeGRXS14* responds to various abiotic challenges, we evaluated the temporal and spatial expression of *LeGRXS14* in tomatoes under different stresses. We generated OE lines to investigate the role of *LeGRXS14* under salt stress and to determine the exact subcellular localization of LeGRXS14 and its expression in different organs. Furthermore, we established the expression profile of genes associated with salt stress in OE lines to investigate the molecular mechanisms underlying the response of *LeGRXS14* to salt stress.

## 2. Results

### 2.1. Phylogenetic Analysis and Chromosomal Distribution of CGFS-Type GRX Proteins

CGFS-type GRXs have four members in tomatoes, namely, LeGRXS14/15/16/17. To identify homologues in the model plant *A. thaliana* as well as in other commercial crops such as *Brassica napus*, *Capsicum annuum*, *Citrullus lanatus*, *Cucumis sativus*, *Glycine max*, *Oryza sativa,* and *Zea mays*, BLASTp searches were performed using the full-length protein sequences of LeGRXS14/15/16/17 as individual queries. The unrooted phylogenetic tree showed that the CGFS-type GRXs could be divided into four major groups (Figure 1a), with each group consisting of nine proteins. LeGRXS14 in the GRXS14 group (highlighted in purple in Figure 1a) showed a close relationship with CaGRXS14, OsGRXS14, and ZmGRXS14, indicating high homology in *C. annuum*, *O. sativa*, and *Z. mays*. The GRXS15/16/17 groups all had high similarity with LeGRXS14. LeGRXS15/16/17 were also most closely related to *C. annuum*, and they had high homology with *G. max* as well; this was in contrast to *O. sativa* and *Z. mays* in the GRXS14 group. We constructed a chromosomal distribution map of CGFS-type GRXs on the tomato chromosomes to understand their genomic distribution, and the results indicated that they were located on different chromosomes. *LeGRXS14* and *17* were on Chr 2, and *LeGRXS15* and *16* were on Chr 6 and Chr 9, respectively (Figure 1b).

### 2.2. Multiple Sequence Alignment and Motif Analysis of GRXS14 in Tomatoes

Peptide alignment performed by the Clustal W program indicated that GRXs from different species showed obvious similarity, with the N-terminus in particular being highly conserved across species. The red boxes and black stars are marked with the active-site motif CGFS (Figure 2a). Considering that motifs are frequently divergent across multiple gene families, we examined the motifs of the GRXs. The MEME web server was used to assess the patterns, and six major motifs were identified. Motifs 1, 2, and 3 were present in all nine GRXS14s. LeGRXS14 clustered with CaGRXS14, OsGRXS14, and ZmGRXS14, while only ClGRXS14 and CaGRXS14 contained motif 5/6 in this cluster (Figure 2b). Appendix A lists the lengths and preserved sequences of each motif in GRXS14. Motifs 1 and 2 have been identified to encode a GRX_PICOT-like domain, while motif 2 contains the active site motif CGFS. Other motifs have not been functionally annotated. 

### 2.3. Expression Patterns of LeGRXS14 Genes in Tomato under Abiotic Stress

To investigate the gene expression level of *LeGRXS14* in different tomato organs, RNA from the root (Rt), stem (St), leaf (L), flower (F), and green fruit (Gf) was extracted for qRT-PCR analysis. The findings demonstrated that *LeGRXS14* was expressed across all of the aforementioned organs. Utilizing the root’s expression level as the control with a relative level of 1, our analysis revealed that the leaf exhibited the highest expression level of *LeGRXS14* at 30.5, followed by the flower, fruit, and stem at levels of 10.7, 8.6, and 4.9, respectively. This finding serves to demonstrate the wide distribution of *LeGRXS14* expression throughout the tomato plant, with particular prominence within the shoots (Figure 3a).

To gain a comprehensive understanding of the potential functions of *LeGRXS14*, we conducted a study that examined the impact of various abiotic stresses on the expression patterns of this gene in tomatoes. The selected stresses included heat, osmotic stress, cold, and salt. The tomato seedlings, aged one month, were subjected to each stress for a period of 24 h. The heatmap revealed that osmotic and salt stress had a bigger impact on *LeGRXS14* gene expression than heat and cold stress (Figure 3b). The expression levels of *LeGRXS14* did not exhibit significant changes under heat stress but did change dramatically under salt and osmotic stress (Figure 3c–f). The highest expression level appeared at 6 h under salt stress, and the gene expression level of *LeGRXS14* increased by 6.2 times compared to that of the untreated seedlings (Figure 3d). In addition, osmotic stress caused an 8.7-fold increase in *LeGRXS14* gene expression in 30 min (Figure 3c). These results suggest that salt and osmotic stress treatments increased the expression of *LeGRXS14* and that osmotic stress enhanced the expression of *LeGRXS14* much more rapidly than the salt stress treatment.

### 2.4. Subcellular Localization of LeGRXS14-GFP

We identified the subcellular localization of the LeGRXS14 proteins to comprehensively evaluate the physiological functions of *LeGRXS14* in *A. thaliana*. In transgenic *A. thaliana* root cells containing *p35S::LeGRXS14-GFP*, GFP fluorescence was detected at high levels in the plasma membrane and nucleus (Appendix A). To verify plasma membrane localization, we further verified plasma membrane localization of the LeGRXS14 protein in plasmolyzed cells and co-localization of LeGRXS14 with a plasma membrane marker (*mCherry-Plasma membrane*) in transiently transformed tobacco (*Nicotiana benthamiana*) leaves. The fluorescence results clearly demonstrate that LeGRXS14-GFP was expressed on the plasma membrane (Appendix A). We also detected co-localization of LeGRXS14 with a nucleus marker (*mCherry-Nucleus*) (Figure 4b), and overlapping pore patterns were observed in tobacco leaves transiently transformed with *p35S::LeGRXS14-GFP* and *mCherry-Nucleus* transgenes. In transgenic *A. thaliana* leaf protoplasts containing *p35S::LeGRXS14-GFP*, the presence of LeGRXS14-GFP was evident within chloroplasts. (Appendix A). Overall, these findings confirmed that LeGRXS14-GFP was localized on both the plasma membrane, nucleus, and chloroplasts. 

### 2.5. LeGRXS14 Overexpression in Response to Salt Stress in A. thaliana

To explore the molecular mechanism governing the response of LeGRXS14 to abiotic stress, we constructed OE lines to investigate their ability to tolerate long-term stress (Figure 5a). Under conditions of low Ca^2+^, both the WT and OE lines showed similar root lengths in ^1^/_2_ MS media without NaCl. However, after 12 days of salt treatment, the OE lines exhibited a reduced root length compared with the WT. Comprehensive statistical analysis of the data distribution clearly showed that the OE lines manifested heightened sensitivity to salt stress in comparison to the WT (Figure 5b). However, no discernible distinction could be observed between the WT and OE lines in a normal Ca^2+^ environment (Appendix A). The results indicated that the upregulation of *LeGRXS14* in *A. thaliana* had an impact on the plant’s reaction to salt stress.

### 2.6. LeGRXS14 Is Related to Plant Resistance to Salt Stress

To gain insight into the molecular responses of *LeGRXS14* to salt stress, various molecular markers were used to examine the relationship between *LeGRXS14* and other molecular elements of plant salt resistance (Appendix A). Quantitative RT-PCR was conducted to verify the gene expression patterns in *A. thaliana* under salt stress. We found that the transcriptional abundance of the salt stress response gene *SOS3* in the OE seedlings was significantly decreased under salt stress (Figure 6k), and the same was observed with the ROS-related factors *RBOHD* and *ZAT12* (Figure 6b,h). The results showed that the overexpression of *LeGRXS14* in *A. thaliana* decreased the expression of salt stress response genes and ROS-related factor genes. In addition, we detected several changes in the transcription levels of salt stress-related genes such as *HKT1* (Figure 6c), *NHX1* (Figure 6d), and *NHX6* (Figure 6g), and the mRNA levels of these were downregulated. These findings suggested that *LeGRXS14* affects resistance to salt stress by inhibiting signaling pathways under salt stress.

## 3. Discussion

GRXs were first discovered by Holmgren in the TRX gene deletion of *E. coli* [7]. They are a class of sulfhydryl-disulfide bond oxidoreductases ubiquitous in prokaryotes and eukaryotes and have a disulfhydryl catalytic center that can control the redox state of target protein activities [19]. CGFS-type GRXs are involved in iron–sulfur cluster assembly, which is required for plant responses to environmental stimuli [5]. The function of CGFS-type GRXs in the response of plants to biotic and abiotic stress has been revealed in earlier studies. It was previously shown that transgenic rice constitutively expressing *OsGRXS15* exhibited enhanced disease resistance to the pathogens *Xanthomonas oryzae pv. oryzae* and *Fusarium fujikuroi* [20]. The protein AtGRXS17 plays a specific role in enabling plants to tolerate moderately high temperatures, and its overexpression demonstrated a protective effect on root meristematic cells by preventing cell death caused by heat stress [21]. The CRISPR/Cas9-based knockout tomato mutant *legrxs16* showed reduced drought resistance compared with the WT, while the overexpression of *LeGRXS16* in soil-grown *A. thaliana* enhanced resistance to oxidative, drought, and salt stresses [22]. Similarly, *legrxs14* displayed a higher degree of susceptibility to drought stress than the WT [18].

As ROS-scavenging network components, CGFS-type GRXs have been identified as being involved in plant resistance to a range of biotic and abiotic stresses. As indicated in the phylogenetic analysis, tomato proteins, including LeGRXS14/15/16/17 and CGFS-type GRX proteins of other species, are clustered into four groups. The differences between these four groups suggest that these proteins underwent significant genetic variation following divergent evolution (Figure 1a). We also found that the N-terminus of GRXS14 is highly and structurally conserved (Figure 2a). It has been reported that the N-terminal active site cysteine residue is crucial for homodimer binding [23]. LeGRXS14 shares a high degree of similarity with GRXs in *C. annuum*, *O. sativa*, and *Z. mays* (Figure 2b). It was found that CC-type *OsGRX8* conferred decreased sensitivity to hormone stress in *A. thaliana* [6], CPYC-type *OsGRX20* conferred increased tolerance to multiple stresses in rice [24], and *ZmGRXCC14* was found to have a correlation with drought stress during the seedling stage [25], suggesting that *LeGRXS14* may also play an important role in plant resistance to various stresses. The function of these genes under biotic and abiotic stresses should be further explored in the future.

We found that the expression of *LeGRXS14* was apparent within the leaf (Figure 3a), and the protein was located on the plasma membrane, nucleus, and chloroplasts (Appendix A and Figure 4). The *LeGRXS14* gene responded to osmotic stress in 30 min, which was faster than the response to salt stress (Figure 3c,d). A previous study showed an increase in drought resistance in the CRISPR-cas9 mutant *legrxs14* [18], and we confirmed that OE seedlings showed reduced salinity resistance (Figure 5a). From this, we speculated that *LeGRXS14* is indeed related to salt stress; however, we did not detect any significant difference between the WT and OE lines under osmotic stress (Appendix A). Under conditions of low Ca^2+^, the overexpression of *LeGRXS14* in *A. thaliana* plants reduced plant resistance to salt stress (Figure 5). We presumed that *LeGRXS14* plays a vital role in the salt stress response, possibly by acting as a negative regulatory factor in salinity resistance, especially in Na^+^ effects. This may be related to the calcium signaling pathway under salt stress. However, the molecular mechanism of the salt stress response of *LeGRXS14* remains unclear.

The salinity resistance markers *SOS1*, *SOS2*, *SOS3*, *ZAT12*, *RBOHD*, *NHX1*, *NHX2*, *NHX5,* and *NHX6* participate in the process of plant salinity resistance [26,27,28,29]. We found that the mRNA levels exhibited obvious changes among these gene markers in the OE lines (Figure 6), particularly *SOS3*, *ZAT12*, and NHXs, for which the gene transcription levels were decreased. *SOS1*, *SOS2*, and *SOS3* encode regulatory components controlling plant Na^+^/H^+^-exchange activity under salt tolerance, and *SOS3* activates and directs *SOS2* to the plasma membrane for the stimulatory phosphorylation of the Na^+^ transporter *SOS1* [30,31]. Thus, the inhibition of *SOS3* affects the phosphorylation of *SOS1* and impairs plant Na^+^/H^+^-exchange activity (Figure 6k). Studies have suggested that NHX exchangers are involved in vacuolar Na^+^ sequestration under salt stress [32,33], and decreased *NHX1* and *NHX6* affect Na^+^ sequestration in the vacuole (Figure 6d,g). External NaCl treatments cause the Na^+^ content to increase in cells, but the exclusion of Na^+^ becomes impaired, leading to Na^+^ accumulation in the cells and ionic toxicity. Salt stress triggers secondary oxidative stress, and previous research has demonstrated that *ZAT12* expression is activated at the transcriptional level during different abiotic stresses [34]. Furthermore, *ZAT12* was thought to be involved in oxidative stress signaling as a negative regulator of Fe acquisition in *A. thaliana* [35]. *PeZAT12* can regulate the expression of *PeAPX2,* thereby scavenging ROS under salt stress in poplar [36]. This may explain why the OE lines were more salt-sensitive than the WT.

In summary, *LeGRXS14* may act as a negative regulatory factor for plant resistance to salt stress by suppressing vital genes such as *SOS3* and NHXs that are crucial for salt stress tolerance, leading to the accumulation of Na^+^ in plant cells. Moreover, *LeGRXS14* overexpression results in the inhibition of *ZAT12,* which impedes the elimination of reactive oxygen species (ROS) during salt stress. As a consequence, *LeGRXS14* negatively influences the salt stress-related genes in *A. thaliana*, reducing the plant’s ability to cope with salt stress. In conclusion, the tomato gene *LeGRXS14* reduces salt stress tolerance in *A. thaliana*.

## 4. Materials and Methods

### 4.1. Plant Materials and Growth Conditions

*Arabidopsis thaliana* seeds underwent sterilization by means of 2.5% PPM (Plant Preservative Mixture; Caisson Labs) for a duration of 3 days at 4 °C. Subsequently, the seeds were placed in 150 mm × 15 mm Petri dishes containing a culture medium comprised of 1/2 Murashige and Skoog salts (MS; Sigma, St. Louis, MO, USA), 1.5% (*w*/*v*) sucrose (Sigma), and 0.6% (*w*/*v*) agar (Sigma) in a pH 5.9 environment. The plants were subjected to growth conditions of ∼110 µmol m^−2^ s^−1^ white light as the photo fluency rate and 16 h of light and 8 h of darkness per day as the photoperiods, while being kept at a consistent temperature of 22 ± 2 °C.

Tomato seeds (‘Micro-Tom’) were sterilized using a combination of 70% ethyl alcohol and 4.5% sodium hypochlorite before being cultivated in Petri dishes. After seven days of growth in an MS medium, the seeds were then transferred to soil, where they continued to be cultivated under controlled conditions. Maintenance of the environmental rooms saw a consistent temperature of 25 ± 2 °C, with a photo fluency rate of white light of approximately ∼110 µmol m^−2^ s^−1^, and the photoperiod included 16 h light/8 h dark cycles. Four-leafed tomato seedlings were treated with drought stress [20% polyethylene glycol (PEG) 6000], salt stress (100 mmol·L^–1^ NaCl), heat stress (40 °C), and chilling stress (14 °C). The sampling time points were 0, 10, 20, and 30 min, and 1, 2, 3, 6, 12, and 24 h after the start of the stress treatment. Following treatment, plants were snap-frozen in liquid nitrogen and refrigerated at –80 °C.

### 4.2. DNA Constructs and Transgenic Lines

The *LeGRXS14* accession numbers for the Solyc02g082200.3 full-length complementary region were obtained from tomato genomes (https://plants.ensembl.org/index.html, accessed on 5 January 2017). The method of Gateway cloning was employed for the construction of *p35S::GRXS14* and *p35S::GRXS14-GFP*. PCR amplification of DNA fragments was carried out from cDNA, followed by cloning into the pENTR vector (Invitrogen). Subsequently, coding sequences from the entry clones were transferred to gateway-compatible destination vectors. The generation of transgenic *A. thaliana* plants was established through *Agrobacterium*-mediated transformation using homozygous transgenic T3 lines that contained a single insertion.

### 4.3. PCR Primers and Vectors

Cloning primers: *GRXS14-cDNA_Fw*, *5′*-ATGTCGCTCGGAATTTCGC;

*GRXS14-cDNA_Rev*, *5′*-TCAGGAGCACAATGTCCT, *GRXS14-cDNA (no stop code) _Rev*, *5′*-GGAGCACAATGTCCTTTCT; vectors: *p35S::GRXS14:pGWB402Ω*, *p35S::GRXS14-GFP:pGWB405*.

### 4.4. RNA Extraction and qRT-PCR

RNA extraction was completed with a Total RNA Kit (Tiangen, Tianjin, China), with 100 mg of each section handled separately. Subsequently, 1 microgram of RNA was used for reverse transcription, followed by the use of the FastKing RT Kit (Tiangen) for cDNA synthesis. The quantitative real-time (qRT) PCR reaction was performed with the Real Universal Color PreMix (Tiangen) and involved an eight-well optical PCR plate for each reaction. Each reaction comprised 7.5 µL premix, 0.75 µL forward primer, 0.75 µL reverse primer, and 6 µL cDNA template. A set of PCR cycle parameters was initiated, which included one cycle of 30 s at 95 °C and 40 cycles of 10 s at 95 °C and 30 s at 60 °C. To ensure that gene expression levels were normalized, actin was utilized as an internal control. Appendix A lists the primers used in this study. A melting curve was created after the 40 cycles to confirm that only one amplified product was obtained. For statistical analysis, the standard *t*-test was used.

### 4.5. GFP Subcellular Localization Analysis

To investigate GRXS14-GFP in *Arabidopsis* seedlings, we generated transgenic plants with 35S promoter-driven GRXS14 (*p35S::GRXS14-GFP*, full-length complementary DNA). To carry out confocal imaging, seven-day-old seedlings grown on ½ MS medium in Petri dishes were studied using a Zeiss LSM 710 microscope. Blue excitation (typically 488 nm) was used for the excitation of GFP. For confocal imaging, a ×20 water immersion objective was used. Our examination included more than 10 independent lines, and we induced plasmolysis using 0.8 M sorbitol. Protoplasts were generated through the enzymatic digestion of plant material using a solution consisting of 20 mM MES (pH 5.7), 1.5% cellulase R-10, 0.4% macerozyme R-10, 0.4 M mannitol, 20 mM KCl, 10 mM CaCl_2_, 0.1% BSA, and during the incubation process.

To study the localization of proteins in the cell, *Agrobacterium tumefaciens*-mediated transient expression in *N. benthamiana* was used to detect the co-localization of LeGRXS14 with a plasma membrane marker (*mCherry-plasma membrane*) and a nucleus marker (*mCherry-Nucleus*) [37]. Briefly, we prepared Luria-Bertani (LB) liquid medium with the appropriate antibiotic mixture for overnight incubation of *Agrobacterium tumefaciens* GV3101 transformed with the target gene and silencing suppressor strain 19K at 28 °C, collected the cultures, and resuspended them with transformation solution (10 mM MES/KOH (pH 5.6), 10 Mm MgCl_2_, and 150 μM acetosyringone), so that the OD_600_ was adjusted by 0.5. We mixed equal volumes of 19K, LeGRXS14-GFP, and markers separately and left this for 3 h before injecting it into one-month-old tobacco leaves. A Zeiss LSM 710 microscope was used to observe the co-localization. For confocal imaging, a ×20 water immersion objective was used. The excitation and emission wavelengths of the GFP signal were 488 nm and 493–570 nm; for the mCherry signal, they were 561 nm and 600–650 nm for the chlorophyll autofluorescence, they were 488 nm and 650–700 nm.

### 4.6. Growth Responses to Salt Stress

For the plant physiology assay, we sterilized WT and OE seeds with PPM and maintained them in darkness at 4 °C for three days. Immediately after this period, the seeds were placed in a low-calcium (0.2 mM CaCl_2_) ½ MS agar medium, with or without 60 mM NaCl. The ½ MS agar medium comprised the vital components of major salts (NH_4_NO_3_, MgSO_4_, KH_2_PO_4_, and KNO_3_; Sigma), a vitamin solution (Sigma), 0.5% sucrose (Sigma), and 0.6% agar (Sigma), with the calcium content increased to 0.2 mM CaCl_2_. Following a growth period of 12 days, we evaluated the seedling root lengths as part of our analysis.

The seeds were also placed in a normal-calcium (2.5 mM CaCl_2_) ½ MS agar medium, with or without 60 mM NaCl. Following a growth period of 8 days, we evaluated seedling root lengths as part of our analysis. 

## Figures and Tables

**Figure 1 plants-12-02320-f001:**
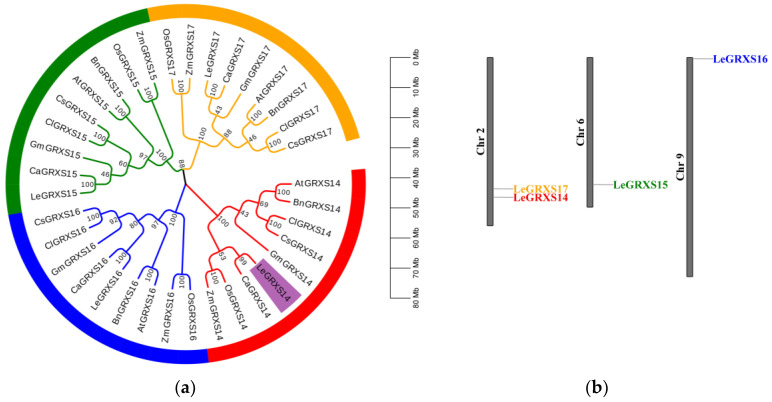
Schemes follow another format. If there are multiple panels, they should be listed as Phylogenetic analysis and chromosomal distribution of CGFS-type GRX proteins: (**a**) The full-length amino acid sequences of CGFS-type GRX proteins were aligned by Clustal W, and the phylogenetic tree was built using the neighbor-joining (NJ) method in MEGA (v11.0). The grouping of the GRX proteins is indicated by different colors, with LeGRXS14 marked in purple. The proteins are abbreviated as follows: *A. thaliana*, AtGRXSx; *B. napus*, BnGRXSx; *C. annuum*, CaGRXSx; *C. lanatus*, ClGRXSx; *C. sativus*, CsGRXSx; *G. max*, GmGRXSx; *O. sativa*, OsGRXSx; *L. esculentum*, LeGRXSx; and *Z. mays*, ZmGRXSx. Appendix A lists the gene IDs. (**b**) Chromosomal distribution map of CGFS-type GRXs on tomato chromosomes.

**Figure 2 plants-12-02320-f002:**
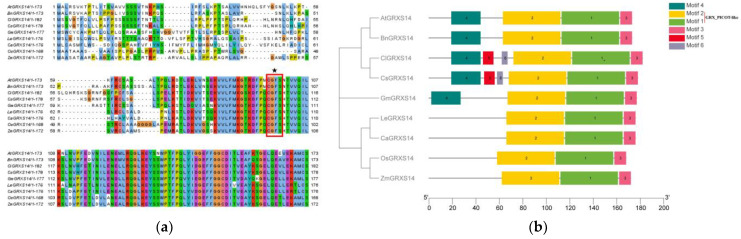
Alignment and motif analysis of LeGRXS14 and its homologous protein sequences: (**a**) Alignment of the tomato protein LeGRXS14 and its homologous protein sequences; (**b**) phylogenetic relationships and conserved motifs of LeGRXS14 and its homologous protein sequences in *A. thaliana*, *B. napus*, *C. annuum*, *C. lanatus*, *C. sativus*, *G. max*, *O. sativa*, *L. esculentum*, and *Z. mays*. Amino acids that are conserved throughout are shaded in different colors. Each motif is represented by a number in a colored box.

**Figure 3 plants-12-02320-f003:**
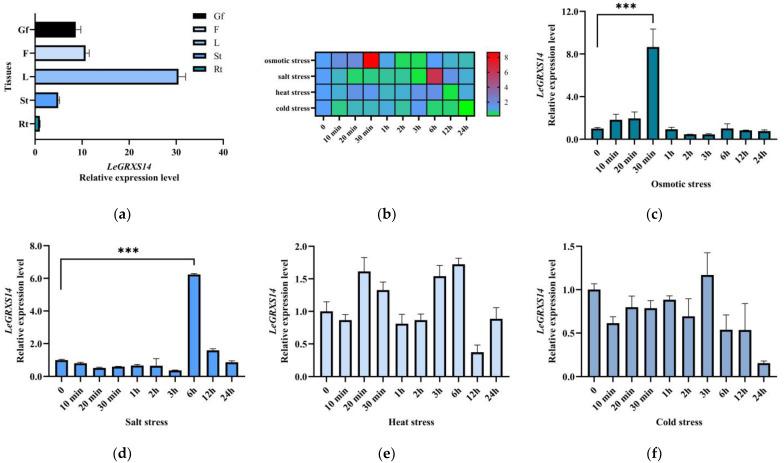
Expression patterns of *LeGRXS14* in tomatoes: (**a**) The expression of *LeGRXS14* in different organs was analyzed using quantitative RT-PCR. Transcripts of *LeGRXS14* accumulate in different organs. The root (Rt) expression data were normalized to 1 and used as a reference to determine the relative *LeGRXS14* expression levels in all other organs. The data are the means ± SD of three independent experiments. Gf-green fruit; F-flower; L-leaf; St-stem; Rt-root; (**b**) heatmap of *LeGRXS14* expression in response to heat, salt, and osmotic stress; (**c**) statistical analysis of *LeGRXS14* expression in response to osmotic stress; (**d**) statistical analysis of *LeGRXS14* expression under salt stress; (**e**) statistical analysis of *LeGRXS14* expression under heat stress; (**f**) statistical analysis of *LeGRXS14* expression under cold stress. Quantitative RT-PCR analysis of *LeGRXS14* transcripts in one-month-old seedlings treated with heat, salt, and osmotic stress. The data in the figure are the mean ± SD (*n* = 4); *** *p* < 0.001.

**Figure 4 plants-12-02320-f004:**
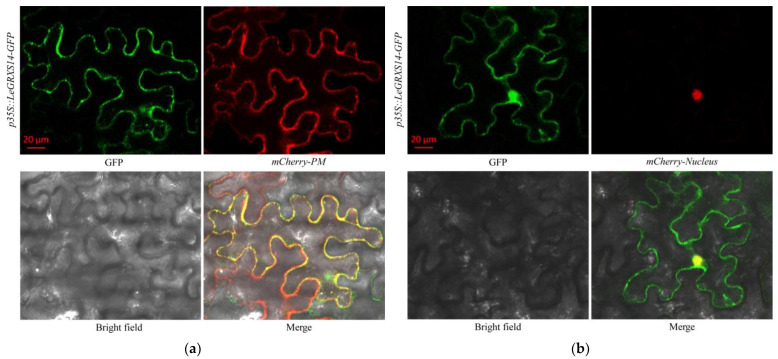
Subcellular localization of LeGRXS14 proteins: (**a**) Subcellular localization of LeGRXS14-GFP and mCherry-plasma membrane in tobacco. (**b**) Subcellular localization of LeGRXS14-GFP and mCherry-nucleus in tobacco. Green shows GFP signals, and red shows mCherry signals. Bars = 20 µm.

**Figure 5 plants-12-02320-f005:**
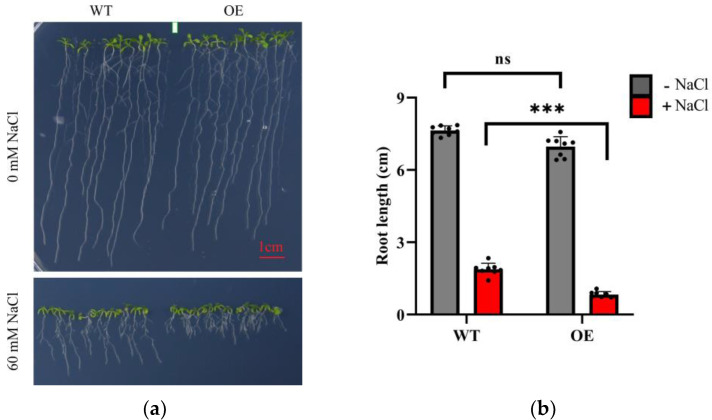
Phenotypic analyses of WT and OE lines under salt stress: (**a**) The WT and OE lines were cultured for a 12-day period on media with or without 60 mM NaCl. (**b**) The quantified data presented in the figure depicts the root length of the WT and OE seedlings. The data displayed in the figure are presented as the mean ± SD (*n* = 8); *** *p* < 0.001.

**Figure 6 plants-12-02320-f006:**
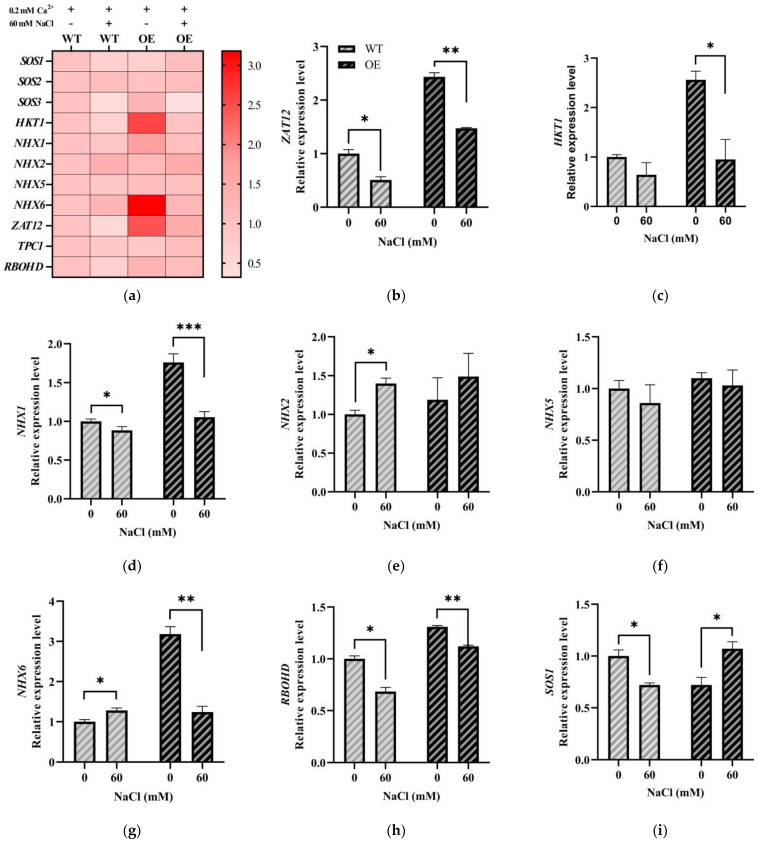
Expression patterns of salt stress-related molecular markers and genes in *A. thaliana:* (**a**) Heatmap expression analysis of molecular marker genes in response to salt stress by qRT-PCR. (**b**) Statistical analysis of *ZAT12* expression in WT and OE under salt stress. (**c**) Statistical analysis of *HKT1* expression in WT and OE under salt stress. (**d**) Statistical analysis of *NHX1* expression in WT and OE under salt stress. (**e**) Statistical analysis of *NHX2* expression in WT and OE under salt stress. (**f**) Statistical analysis of *NHX5* expression in WT and OE under salt stress. (**g**) Statistical analysis of *NHX6* expression in WT and OE under salt stress. (**h**) Statistical analysis of *RBOHD* expression in WT and OE under salt stress. (**i**) Statistical analysis of *SOS1* expression of WT and OE under salt stress. (**j**) Statistical analysis of *SOS2* expression in WT and OE under salt stress. (**k**) Statistical analysis of *SOS3* expression in WT and OE under salt stress. (**l**) Statistical analysis of *TPC1* expression in WT and OE under salt stress. Quantitative RT-PCR analysis of *LeGRXS14* transcription in 10-day-old seedlings subjected to 60 mM NaCl treatment. The data displayed in the figure are presented as the mean ± SD (*n* = 4); * *p* < 0.05, ** *p* < 0.01, *** *p* < 0.001.

## Data Availability

Not applicable.

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
