# Peer review of "LeGRXS14 Reduces Salt Stress Tolerance in Arabidopsis thaliana"

_plants, 2023, doi:10.3390/plants12122320_

Round 1
Reviewer 1 Report
The authors have studied the LeGRXS14 from tomato and heterologously used the Arabidopsis system for salt stress.
Major Comments:
1. Line 63: Are these OE lines in a tomato background?
2. Line 77: The authors have chosen all commercial crop plants from the evolutionary closure family for phylogenetic analysis. Is it possible to use divergent groups?
3. Line 136: What is the significance of transient upregulation of LeGRXS14 in response to osmotic stress? How many biological and technical replicates were performed? What reference genes were used for the normalization of qRT PCR data?
4. Line 156: To confirm plasma membrane (PM) localization, could you please show the colocalization of LeGRXS14 with PM-marker (e.g., FM64 dye or any PM-localized protein) data?
5. Line 167: How many independent OE lines were analyzed? In supplementary figure S1, OE lines do not show any sensitive phenotype. Could authors justify it?
6. Line 169: Why did you expect higher expression levels of LeGRXS14 in OE lines in response to salt treatment? Isn’t the 35S promoter constitutively overexpressing the gene?
7. Line 181: What age were Arabidopsis plants used for qRT-PCR sample preparation? What time points were chosen for gene expression analysis and why?
8. Line 183: SOS3, RBOHD, and ZAT12 expression is also significantly downregulated in WT plants. Then what is the role of LeGRXS14?
9. Line 345: What confocal settings were used?
Minor Comments:
1. Line 107: What are these motifs and their significance?
2. Line 164: Why used low Ca2+ levels?
English correction is needed for clear message transfer.
Author Response
Dear Professor,
Thank you very much for your patience in reviewing this article. We have made up some experiments, which are summarized in this word, and we invite you to consult them.
Thank you!

Reviewer 2 Report
The manuscript LeGRXS14 Reduces Salt Stress Tolerance in Arabidopsis thaliana by Lulu Liu, Xiaofei Li, Mengke Su, Jiaping Shi, Qing Zhang, Xunyan Liu presents an experimental-theoretical presentation of the results of the LeGRXS study of transgenic Arabidopsis plants. The manuscript contains all the necessary parts, except for the conclusion. There is no design in the manuscript in accordance with the rules, fonts and highlighted fragments of captions for figures do not differ from the main text.
Although this work is of some interest, the terminology used and the final seme contain unacceptable errors. These inaccuracies and confusion are so significant that I was faced with the question of the inadmissibility of this publication. However, seeing that the misconceptions relate to the fundamental foundations of beology and could be made by a person who did not complete the educational cycle, I believe that one can try to provide the authors with the opportunity to correct. If the manuscript is not corrected, it can by no means be printed in this form. It remains to be hoped that these "inaccuracies" are accidental.
Authors should carefully read the terminology. Roots, flowers, leaves, etc. are not tissues, but are complex organs consisting of several tissues. For example, the root includes (rather roughly) the epidermis, cortex (parenima), recycle, endoderm, xylem, phloem, and parenima tissues of the central cylinder. It should be taken into account that the ploidy of the cells of these tissues is different and the measurement of expression in this case can be fantasy without additional measures.
In addition, I remind you that these cells have different functionality and differ significantly in gene expression.
The authors' indication of where the product is localized (enzymes are proteins) does not inspire confidence ... You can start with the fact that the cell walls and probably the nucleus glow, although it will not be possible to say with certainty, it is possible that this is a site with plasmolysis. But how to explain that the pictures are mixed up?
The statement about localization is surprising in that we see only epidermal cells, and although they are root cells, it is not clear whether this was worth pointing out. At the same time, the authors talk about plasmolysis, but they do not find it possible to indicate exactly where they see it with an arrow. The statement that "LeGRXS14-GFP was expressed on the plasma membrane" confuses the reader. Does the expression of genes, in their opinion, really occur on the membrane, and not in the nucleus?
The entire text of the manuscript describes measurements in organs, which the authors call tissues, so the last "icing on the cake" becomes completely expected.
Apparently trying to understand the intricacies of the mutual influence of the studied genes, the authors decided to draw a diagram that can really open a new chapter in cell biology, and this is frightening.
Dear authors, please note that the lipid membrane also exists around the vacuole, and as for the nucleus, it is still more complicated there, the location of APH and the role of this enzyme in the production of reactive oxygen species should be clarified.
The only thing that makes the authors leave a chance to correct this manuscript is curiosity and the hope that after all this is some kind of crazy, randomly sent draft and the authors will quickly fix it.
Author Response

(The authors gave the same response as above.)

Round 2
Reviewer 1 Report
Point no.5 Could the authors explain in detail in which condition OE lines showed sensitive phenotype, as shown in Fig. 5a. In the author's response to point 5, the authors have shared pictures of OE lines showing no sensitive phenotype towards salt. Could the authors exactly mention the conditions used in this experiment?
Point No. 6 I didn't question the result but asked why in 35S promoter is responding the stress.
Most of the comments addressed here are not very clear.
It is improved a lot but may need some minor adjustments.
Author Response
Dear Professor,
Thank you for arranging a timely review of our manuscript. Your constructive comments and valuable suggestions have helped us improve our research's quality.
We have thoroughly revised the manuscript according to your comments and suggestions, and we have engaged the services of MDPI for professional language polishing and editing. In addition, we have provided detailed responses to each of your comments in the attached document. We hope that you will find the revisions satisfactory.
We appreciate your critical evaluation and insightful feedback, which have greatly contributed to the enhancement of our research. Your guidance has been invaluable, and we are thankful for your efforts.
Thank you once again for your time and consideration. We look forward to your continued support and guidance in the future.
Best regards,
Lulu Liu

Reviewer 2 Report
Dear authors, please note that the inscriptions on many of the figures are illegible - this should be corrected.
Dear authors, please pay attention to Figure 4a, where you need to rearrange the pictures swap the center one on the place "merge" .
Author Response
Dear Professor,
Thank you for arranging a timely review for our manuscript. Your constructive comments and valuable suggestions have helped us improve our research's quality.
We have thoroughly revised the manuscript according to your comments and suggestions, and we have engaged the services of MDPI for professional language polishing and editing.
We appreciate your critical evaluation and insightful feedback, which have greatly contributed to the enhancement of our research. Your guidance has been invaluable, and we are thankful for your efforts.
Thank you once again for your time and consideration. We look forward to your continued support and guidance in the future.
Best regards,
Lulu Liu
